# Mutual Interactions Between Microbiota and the Human Immune System During the First 1000 Days of Life

**DOI:** 10.3390/biology14030299

**Published:** 2025-03-16

**Authors:** Muy Heang Tang, Ishbel Ligthart, Samuel Varga, Sarah Lebeer, Frans J. van Overveld, Ger T. Rijkers

**Affiliations:** 1Department of Science and Engineering, University College Roosevelt, 4331 CB Middelburg, The Netherlands; m.tang@ucr.nl (M.H.T.); i.ligthart@ucr.nl (I.L.); s.varga@ucr.nl (S.V.); f.vanoverveld@ucr.nl (F.J.v.O.); 2Lab of Applied Microbiology and Biotechnology, Department of Bioscience Engineering, University of Antwerp, 2020 Antwerpen, Belgium; sarah.lebeer@uantwerpen.be

**Keywords:** gut microbiota, neonatal immune system, bacterial colonization, mode of delivery, prenatal delivery, short-chain fatty acids (SCFAs)

## Abstract

Human embryonic and fetal development takes place in the sterile environment of the uterus. The first lymphoid progenitor cells can be detected as early as week 4 of gestation, and by week 20, all components of the immune system have, in fact, been formed. Because of the sterile environment of the uterus, the neonatal immune system is not challenged by microbial antigens until birth. The moment the newborn is exposed to the outside world, it comes into contact with a vast array of micro-organisms, including potential pathogens. During the first 1000 days of life, the neonatal immune system is activated by antigens of microbial and non-microbial origin, and it will respond to these triggers and develop immunological memory. In order for the immune system to be able to mount a specific response to microbial antigens (“non-self”), it must be able to make a distinction between “self” and “non-self”. During development and differentiation of lymphocytes, the lymphocytes with a specificity for “self” antigens become inactivated (tolerance) thus preventing auto-immune diseases. The immune system should also not respond to food proteins nor to commensal bacteria. Most of the commensal bacteria reside in the gut. The mucosal immune system of the gut therefore comes into close contact with food antigens and commensal microbial antigens. These mutual interactions, early in life, shape the balanced development of the neonatal immune system, which is essential to prevent infections, and avoid excessive activation (allergy) and misdirected responses (autoimmunity) later in life. The critical players in these interactions (cells, molecules, and pathways of the immune system, as well as specific microbiota and their metabolites) will be discussed and put in context.

## 1. Introduction

Directly after birth, the human newborn enters a world teeming with micro-organisms. In order to protect itself against pathogenic micro-organisms, the innate and adaptive immune systems develop during fetal life. During postnatal life, the intestines, as well as all other body surfaces exposed to the outside world, become colonized by micro-organisms. While pathogenic micro-organisms should be recognized and eliminated by the immune system, commensal micro-organisms contribute to human health and well-being and therefore should not be attacked. Philippe Sansonetti has termed this situation as war and peace on mucosal surfaces [1]. In this review, we will describe the development of the human immune system during the first 1000 days of life, a period during which the immune system further matures and diversifies, and the mutual interactions with the microbiota, especially those residing in the gastrointestinal (GI) tract [2]. The cellular and molecular details of these interactions will be discussed and put into context.

## 2. Pre- and Postnatal Development of the Immune System

### 2.1. Prenatal Development of the Immune System

The first steps in the development of the immune system have already been taken during early embryonic development. Thanks to large-scale single-cell genomics and imaging techniques, the concurrent development of the hematopoietic and immune systems has been mapped throughout early prenatal life. In the first post conception week (PCW), the primary human yolk sac first appears and is closely associated with the development of the first fetal blood cells (Figure 1) [3]. Up to 3 weeks after fertilization, trophoblast cells arise from the blastocyst, initiating implantation into the uterine wall and early placental development [4]. Around the same time, the main site of hematopoiesis is initiated in the yolk sac around 2 PCWs, before eventually transitioning to the aorta–gonad–mesonephros (AGM) region. By 4 to 8 PCWs, hematopoietic activity gradually shifts to the fetal liver and is concentrated here until 20 PCWs, when hematopoiesis shifts once more to predominantly occur in the fetal bone marrow and spleen, producing all types of blood cells, including the cells of the immune system, and continuing after birth throughout adulthood (Figure 1) [2,5].

#### 2.1.1. Development of Innate Immune System

At 4 PCWs, the yolk sac contains definitive hematopoietic progenitors, macrophages, mast cells, innate lymphoid cells, monocytes, as well as megakaryocytes [6,7,8]. Neutrophils, basophils, and eosinophils, on the other hand, remain absent until hematopoiesis in the bone marrow is established [6,9]. During the sixth PCW, the embryonic pancreas, brain, and placenta are populated with precursors of antigen-presenting cells such as macrophages, microglia, and Hofbauer cells (fetal placental macrophages), respectively. These cells appear before blood cell formation starts in the fetal liver, indicating that macrophages of erythromyeloid progeny from the yolk sac or AGM are able to migrate to populate other tissues [6,10,11,12]. These macrophages are in an activated state and can contribute to immune regulation and phagocytosis halfway through gestation. They begin expressing pro-inflammatory genes (e.g., NKFB1, *FOS*, *JUN*) as early as 8 to 10 PCWs and can exhibit scavenger functions by 12 to 16 PCWs [13,14].

While yolk sac-derived macrophages persist in certain tissues, they are gradually phased out and replaced in other locations, such as the lung, heart, and gut, by monocytes derived from hematopoietic stem cells [15,16]. Monocytes account for approximately 3% of cells in the fetal blood, with their levels increasing linearly with gestational age [13]. These cells are initially detected in the fetal liver but later shift their production site to the bone marrow [5,17]. Although a distinct mast cell population is phenotypically developing in the fetal skin around 12 to 14 PCWs, these cells express minimal levels of detectable granules and a low but maturing expression of the IgE receptor on their surfaces [18,19]. This demonstrates a minimal role in allergic responses but a potential contribution to mucosal immunity and angiogenesis.

Innate lymphoid tissue inducers are cells that aid in forming secondary lymphoid organs [2,6,20], specifically embryonic lymph nodes, by interacting with lymphoid tissue organizer cells [6,21] to generate conditions that facilitate innate lymphocytes to develop very early in the human embryo. At 18–23 PCWs, mucosal-associated invariant T cells (MAIT) can be found in the fetal small intestine, livers and lung [22].

#### 2.1.2. Development of the Humoral Immune System

At 7 PCWs, the earliest signs of the B cell lineage appear, with mature B cells observable by 9 PCWs [6,7]. Although intestinal fetal B cells initially derive from follicular and transitional B cell types, by mid-gestation, the bone marrow becomes the primary source of B cells. By 9 to 12 PCWs, lymphoid progenitor cells differentiate, with some maturing in the thymus and others developing into B cells and natural killer (NK) cells. Moreover, maturing B cells have also been confirmed to exist outside blood vessels in the submucosa of the gut and the dura mater of the meninges [13,23,24].

By 16 PCWs, B cells begin producing immunoglobulins [25], while NK cells become active in innate immune responses. The adaptive immune system becomes more robust as gestation progresses, with B cells increasing immunoglobulin production to strengthen humoral immunity. While fetal B cells attain a progressive increase in B cell receptor diversity from early stages [6,21,25], the formation of germinal centers and hence the specialization of B cells are limited until antigen exposure after birth [6,7]. NK cells play a role in the innate immune response and immune surveillance when mature. A high concentration of fetal NK cells is present in the liver, lungs, and spleen, whereas lower levels are found in the fetal bone marrow and mesenteric lymph nodes [26]. Despite fetal NK cells from the liver and spleen being functionally immature and less active, compared to adult NK cells, they exhibit signs of cytotoxic activity. They are abundant within the infant intestines and show higher degranulation levels than their adult NK cell counterparts [26,27,28]. In the final stages of fetal development, the immune system undergoes fine-tuning and maturation to prepare for exposure to the extrauterine and external environments.

#### 2.1.3. Development of the Cellular Immune System

At 8 weeks, early lymphoid progenitor cells migrate from the fetal liver to the thymus, where they are able to develop, giving rise to naïve T cells (Figure 1) [29]. Prenatally, innate lymphoid cells exist in subtypes. These include cells expressing EOMES and TBX21 (type 1), cells expressing RORC and CCR6 (type 3) (similar to T helper 17 cells and NKT-like cells), and PDCD1-expressing CD8aa cells (similar to CD8aa^+^ T cells). Postnatally, the levels of PDCD1-expressing CD8aa cells have been shown to increase, reaching their peak during childhood [13,30]. Distinct cell types in the thymic microenvironment, such as the stromal cells, thymic epithelial cells, mesenchymal cells, endothelial cells, and other non T lineage immune cells, interact extensively, facilitating the continuous progression from early thymic progenitors to various mature T cell types, promoting T lymphopoiesis [31,32,33]. Following this interaction, by 12 to 14 PCWs, circulating T cells can be observed in the periphery, indicating that the thymus has now developed the ability to produce functional, responsive T cells [34,35]. Studies on neonatal adaptive immunity suggests that the early-life T cell populations are primed to generate tolerogenic responses, characterized by effector T cells with innate-like functions and an increase in peripherally induced regulatory T cells (pTregs) [36,37,38,39,40].

Although dendritic cell (DC) production is initiated and first observed in the fetal liver around 7 PCWs, the potential production of plasmacytoid DCs from both myeloid and lymphoid origins in the yolk sac have been discovered [7]. By 13 PCWs, DC subsets can also be found in the fetal spleen, skin, thymus, and lung [41]. As pregnancy progresses to between 17 and 20 PCWs, splenic DCs mature and gradually take on roles as antigen-presenting cells for T cells [41,42]. Fetal DCs contribute to immune regulation. They induce the differentiation of regulatory T cells (T-regs) and promote T cell IL-4 production, supporting a Th2 response. They also inhibit T cell TNF-α by increasing arginase 2 expression in conventional DCs within the fetal spleen to further aid in the suppression of inflammation [41].

#### 2.1.4. Development of Immunological Memory

The analysis of umbilical cord shows that T cells and B cells in the peripheral blood of human newborns are all virtually in the naïve state (Figure 2) [43]. This makes sense because the intrauterine environment is virtually devoid of antigenic stimuli (see below). Yet, T cells with a memory phenotype have been identified in the fetal intestine, implying the possibility of antigen recognition [6,44,45,46,47]. When studying CD4^+^ T cells in the intestines, memory T cells and T-regs showed signs of proliferation and activation [6,46,47]. During the prenatal stage, ingestion of amniotic fluid in the womb could potentially present microbial antigens (see below), which could lead to the activation of the mucosal immune system of the gut.

### 2.2. Postnatal Development of the Immune System

Directly after birth, the infant immune system is exposed to an overwhelming number of antigenic stimuli, both of microbial nature and otherwise. While their immune system is perfectly capable of responding to these stimuli (with one exception, discussed below), every immune response is a primary response, taking up to 10 days to reach its peak and plateau. Neonates, in the first period after birth, are protected by transplacentally obtained IgG from the mother [48]. Due to the 21-day half-life of IgG, most of the maternal IgG is depleted by 100 days of age, and from this moment onwards, the infant depends on the functionality of its own immune system [49].

During the first 1–2 years of life, children experience many respiratory and GI infections, commonly referred to as childhood diseases [50]. In the Dutch language, the term “childhood diseases” is also used in a non-medical context, namely for inevitable start-up problems, which are overcome by themselves. In English, “teething trouble” is the expression used for these events. The frequency of childhood diseases decreases over time, which reflects the building up of immunological memory, conferring clinical protection against pathogens that have been encountered before.

The exceptions to neonatal immune capabilities/capacities are infections with encapsulated bacteria, such as *Streptococcus pneumoniae* and *Salmonella typhi*. The B-lymphocyte responses to the capsular polysaccharides of these bacteria are absent at birth and cannot be demonstrated until 1.5–2 years of age [51]. Bacterial polysaccharides are so-called T cell-independent antigens, and in order to achieve full B-lymphocyte activation, co-stimulation via complement receptor type 2 (CD21) is necessary [52,53]. B-lymphocytes with high expression of CD21 reside in the marginal zone of the spleen (Figure 3, panel B), and it is at this site where polysaccharide antigens are encountered and a response is initiated [54]. In infants and young children, marginal zone B-lymphocytes lack CD21 expression and thus are unable to respond to polysaccharides [55]. The conjugation of bacterial polysaccharides with carrier proteins (so-called conjugate vaccines) changes the nature of the polysaccharide antigens and overcomes neonatal unresponsiveness [56]. Conjugate vaccines against *S. pneumoniae*, *Haemophilus influenza* type b, and *Neisseria meningitidis* have been successfully implemented in childhood vaccination programs [56].

The intimate interaction between gut microbiota and the developing immune system is also reflected in the occurrence of anti-A and anti-B blood group antibodies. A person with blood group A has anti-B antibodies, while a person with blood group B has anti-A antibodies. Anti-blood group antibodies are generally not detectable before one year of age. Although, from the work of Karl Landsteiner, this phenomenon has already been known for almost a century [57], the underlying mechanism still has not been proven in detail. The current view is that gut microbiota carrying oligosaccharides with strong antigenic similarity with blood group antigens can induce an antibody response. While microbiota with blood group A or B will be present in every individual, the immune system is tolerant for self-antigens and thus will not generate an immune response to their own blood group antigen. Indeed, persons with the blood group AB do not generate blood group antibodies [58]. The microbiota which carries oligosaccharides similar to human blood group antigens have not yet been specified. Apparently, they are not highly pathogenic, because the infection frequency of persons with blood group AB is no different from others. It should be noted that blood group A and AB persons may be more susceptible to smallpox [59].

During the first 500 days of life, T and B cell numbers gradually increase and obtain a mature phenotype [60,61]. Afterwards, gradual changes take place until adulthood. After the physiological dip at 100 days of age, serum IgG levels increase and reach adult levels at 500–1000 days. IgM and IgA levels increase at a slower rate and reach adult levels later [62,63].

## 3. Development of the Infant Gut Microbiome

### 3.1. Maternal Microbiota

#### 3.1.1. The Role of Maternal Microbiota During Fetal Development

The earliest interaction between the immune system of the developing embryo and external microbiota takes place with maternal microbial antigens. The maternal microbiome has been shown to be vital in producing the essential vitamins, amino acids, and other metabolites needed to facilitate the growth and development of the fetal immune system [64]. This finding is supported by the discovery that the maternal microbiome undergoes a dramatic change during pregnancy, establishing a particularly stable bacterial population in the vaginal tract (characterized by a surge in *L. crispatus*, *L. gasseri*, *L. jensenii*, and *Limosilactobacillus vaginalis* populations) and an embryo-nurturing population in the maternal GI tract [65,66]. Additionally, short-chain fatty acids (SCFAs), vital bacterial products in healthy development, are synthesized via fermentation in the maternal gut (from a surge in SCFA-producing bacteria) and play a central role in the establishment of a healthy immune system and metabolic profile within the developing embryo [66,67,68]. This change indicates that bacterial abundance promotes the healthy growth of the developing embryo through metabolite production. Although it is outside the scope of this review, it is interesting to note that maternal microbiota play a role in the fetal neurodevelopment and that Th17 cells are involved in this interaction [69,70,71]. The resident microbiota of the maternal gut and vaginal tract may also play a role in placental colonization. This will be explored further in the subsequent sections. For now, Aagaard et al. characterized a unique placental microbiome composed of symbiotic species, which were compositionally similar to the microbiome of the oral cavity and the upper GI tract [72].

In summary, the unique composition of the maternal microbiome could play a vital role in the healthy development of the fetus through providing a stable input of essential metabolites and preventing the colonization with potentially Th17-mediated inflammation-inducing bacteria. The stability of the maternal vaginal microbiome also plays a role in preventing colonization with pathogenic species which may play a role in adverse pregnancy outcomes (APOs).

#### 3.1.2. The Composition of Maternal Microbiota Can Be Changed Through External Factors

Despite the observed stability of the maternal microbiome, changes in diet, mental state, and medication can result in microbiota alterations, which could have an effect on the developing fetus. The primary modulator of the maternal gut microbiome (and thus the healthy development of the fetus) is the maternal diet, which may be altered during pregnancy. On the large scale, the maternal diet dictates the materials available for the development of the fetus, including the naïve immune system. The examination of maternal malnutrition has shown that a deficiency in vital building blocks such as zinc and vitamins (choline, folate, B2, B6, and B12) result in impaired lymphocyte activity, decreased antibody concentrations, and the dysregulated epigenetic regulation of immune cells [73]. Dramatic changes in the maternal diet have been shown to shift the composition of the maternal gut microbiome, with lower levels of *Bifidobacteria* and *Bacteroides* subspecies and higher levels of *Firmicutes* subspecies being associated with a diet rich in saturated fatty acids and carbohydrates (Figure 4) [74,75]. These changes can influence the immune development of the fetus at two levels. Firstly, at the level of bacterial metabolites such as SCFAs and vitamins, the loss of vital microbiota prevents the fetus from accessing the necessary building blocks to mature (see below). Thorburn et al. fed pregnant mice with a high-fiber diet, promoting SCFA-producing bacteria and lowering the incidence of allergic airway disease in the offspring [76]. Secondly, at the level of colonization, the dysregulation of the maternal microbiome may prevent the fetus from being colonized with vital microbiota in utero, preventing it from developing a healthy immune phenotype (see below for the in utero debate). Gestational exposure to a high-fat diet has been shown to decrease regulatory T cell levels and increase dispositions for autoimmune conditions and pathogenic infection [76,77,78,79,80,81].

The secondary modulator of maternal microbiota is maternal stress. A study by Jašerevič et al. demonstrated that maternal stress during prenatal development modulated the dynamics and composition of the gut and vaginal microbiome. The observed change included a shift toward facultative anaerobe species in the gut (*Mucispirillum* and *Desulfovibrionaceae*) and a disruption in the dynamics of vaginal microbiota diversity (Figure 4) [82]. This change in maternal microbiota then reflected the composition of the neonatal microbiome vertical transmission and may have altered fetal development through the formation of inflammation-associated metabolites such as hydrogen sulfide [82]. Finally, the administration of antibiotic treatment during pregnancy has been shown to disrupt the maternal gut microbiome and affect fetal development [83,84,85,86]. This provides further evidence for the link between the maternal microbiome and prenatal development, as antibiotic use has been associated with multiple APOs (Figure 4) [83]. An additional link between the maternal microbiome and the developing fetus can be seen in the case of infection-based APOs. Intrauterine infections are the most common causes of preterm delivery, accounting for 20% of preterm births and resulting in 450 preventable neonatal deaths per hour worldwide [87,88]. This showcases the strong relation between the composition and dynamics of the maternal microbiome and prenatal development.

### 3.2. Prenatal Bacterial Colonization

There is clear evidence that the maternal gut microbiome influences the development of the fetus; however, there is an ongoing debate on the extent to which the maternal microbiome leads to the bacterial colonization of the fetus before birth. This debate is between the sterile womb hypothesis, whose supporters pose that primary bacterial colonization takes place during and after birth, and the in utero hypothesis, which proposes that the fetus is exposed to microbiota prenatally through the amniotic fluid, placenta, and umbilical cord. The evidence for both hypotheses is based on the bacterial culturing and 16S ribosomal sequencing of both the meconium and the composition of the womb.

#### 3.2.1. The Sterile Womb Hypothesis

The sterile womb hypothesis proposes a view of the womb which is devoid of possible bacterial colonization [89]. This older view became prevalent because of the failure to culture bacterial colonies from the infant meconium [90]. The argument has been strengthened by the outcome of molecular techniques, such as 16S rRNA sequencing, indicating that the meconium does not have a distinct microbiome [91]. Kennedy et al. investigated the fetal meconium through rectal swabs during elective cesarean sections before the application of antibiotics and labor induction. These samples were then compared to fecal samples of infants hours and, days after birth, through 16S rRNA sequencing. The results of this study indicate that no microbial signal was detected in most of the fetal meconium samples. The samples with a detectable biomass were most likely skin contaminants as their profile consisted mostly of *Staphylococcus epidermidis* [92]. Further evidence for the sterile womb hypothesis comes from the taxonomic analysis of gut microbiota in relation to different delivery methods. The microbial composition of neonates delivered vaginally reflect the composition of the maternal vaginal microbiome, with a more homogeneous multi-microbial composition, while neonates delivered through cesarean section were inoculated with the maternal skin microbiome, which is less homogeneous and dominated by particular taxa including *B. licheniformis* [93]. Despite the evidence proposed by this hypothesis, studies into the biomass of the placenta, amniotic fluid, and fetus have provided interesting insights into a possible alternative.

#### 3.2.2. The Sterility of the Placenta, Amniotic Fluid, and Fetus

The primary evidence for a non-sterile womb is the finding of a substantial microbial biomass in the placenta. Under sterile conditions, Aagaard et al. examined the microbiological composition of extracted placental specimens from 320 subjects. These specimens were then compared to the compositions of other human body site niches, including the oral cavity, skin, nasal cavity, and gut from non-pregnant controls. The results of this ribosomal analysis showed that the placenta harbors a unique microbiome composed of multiple phyla, including *Firmicutes*, *Tenericutes*, *Proteobacteria*, *Bacteroidetes*, and *Fusobacteria*. Additionally, the comparative analysis demonstrated that the microbiome of the placenta is compositionally most akin to the oral microbiome of non-pregnant controls. Finally, this study showed that the microbiome in question was associated with a history of antenatal infections [72]. Additional evidence for the colonization of the womb comes from the 16s rRNA sequencing of amniotic fluid samples, which have been found to house a low biomass characterized by low richness and diversity, mostly consisting of *Proteobacteria* [94]. Additionally, past studies have shown that the volume of the amniotic fluid is not constant throughout pregnancy and may be altered by fetal ingestion [95]. When taken together, these ideas contribute to the conclusion that fetal colonization may occur through amniotic ingestion and placental exposure. The sterility of the fetus is particularly disputed and will be explored below.

#### 3.2.3. The in Utero Hypothesis

The in utero hypothesis is the antithesis to the sterile womb hypothesis and poses the idea that fetal colonization takes place before inoculation with vaginal or dermal microbiota during birth. This hypothesis responds to the methodological shortcomings of the sterile womb hypothesis. A particular argument was made that negative culture data do not rule out low-level colonization.

The 16s rRNA sequencing of the umbilical cord blood of healthy infants born by elective cesarean section has shown DNA from commensal bacteria of the *Enterococcus*, *Streptococcus*, *Staphylococcus*, or *Propionibacterium* genus [96]. This suggests a possible mother–child axis of microbial colonization. In their research on rodent models, Jiménez et al. isolated a strain of *E. faecium* and labeled it before maternal oral inoculation. The presence of this labeled strain could be detected in the meconium of the neonatal mice obtained by cesarean section one day before scheduled delivery [97].

Further analysis of the meconium of neonates with DGGE and HITChip methods provided additional evidence of bacterial presence. Specifically, the meconium was found to contain Bacilli and other Firmicutes bacterial species, while fecal samples of the same infants days later were dominated by Proteobacteria [98]. Additional culturing techniques showed that the meconium may be colonized with *Staphylococcus* strains, while fecal samples of older infants contained *Enterococcus*, *Escherichia coli*, *Escherichia fergusonii*, *Klebsiella pneumoniae,* and *Serratia marcescens* [98].

Despite these findings of possible bacterial colonization in utero, methodological issues remain. The environmental contamination of samples is a great risk. A study by Dos Santos et al. demonstrated the ease of false positivity in the analysis of meconium samples, which were not above background signals. Additionally, the bacterial contamination that was confirmed was particularly uncommon and was consistent with post-natal skin colonization [91]. Finally, due to the low biomass of possible microbiota, 16s rRNA sequencing is the prevalent technique for bacterial recognition. However, this sequencing technique is prone to false positives at the hand of dead bacteria whose contents may be recognized. The role of the placenta is particularly debatable in this regard, as the presence of bacterial biomass may not be a marker of fetal colonization but instead of placental function in waste management [90]. These criticisms were exemplified in a study by Hansen et al., where the genetic sequencing was able to recognize bacterial colonization in two thirds of meconium samples analyzed at levels too low to be reliably confirmed by PCR [99]. Therefore, although there may be evidence for an in utero origin to bacterial colonization, the methodology behind these findings may be limited. The current weight of evidence suggests that the healthy human fetus probably does not harbor an established live bacterial community in utero.

### 3.3. Bacterial Colonization of Premature Children

Despite the inconsistencies in the evidence surrounding fetal exposure to microbiota in utero, the maternal and fetal exposure to pathogenic bacteria has been a persistent and observed cause of APOs. Globally, severe infection and preterm birth account for a combined 54% of neonatal deaths [87]. Both factors are associated with pathogenic bacterial exposure and are exceedingly prevalent in low-income countries [87]. In the case of neonatal death, exposure to bacterial biomass or metabolites becomes a critical step in pathogenesis; however, neonates born prematurely because of non-pathogen-related causes are also distinct in their microbial colonization.

#### The Development of the Gut Microbiome Is Specific to Gestational Age

Gestational age (GA) is a determining factor in the colonization and development of the neonatal gut microbiome. Preterm infants have been shown to contain a functionally different gut microbiome compared to full-term infants, with preferential colonization by facultative anaerobe species [100,101]. Studies into differences in the initial microbial composition of preterm infants compared to full-term infants showed predominant Proteobacteria colonization, as well as high levels of *Staphylococcus* and *Haemophilus,* which decreased after delivery in preterm infants [102]. Additional analysis of very preterm infants delivered at GA ≤ 30 weeks showed drastic differences in bacterial strains, with *Staphylococcus* species composing the majority of the fecal biomass. Additionally, very few infants exhibited colonization by *Bifidobacterium*, *Bacteroides*, and *Atopobium* species at birth and during the early periods of development [103]. Thus, preterm infants also differ from full-term infants in their exposure to bacterial metabolites. Concentrations of SCFAs are particularly low in preterm infants due to differences in bacterial colonization, possibly shaping the development of the neonatal immune system (Figure 4) [104].

Due to the instability of the preterm gut microbiome, the administration of antibiotics shortly after birth can be particularly damaging. The deleterious effect of broad-spectrum pre-partum antibiotics in premature infants has been modeled by observing increased colonization by the *Enterobacteriaceae* species after birth. It is important to note that preterm infants are often subject to pathogenic colonization (group B *Streptococci*) as a result of possible bacterial infection leading to APOs. In these cases, antibiotics are a standard part of treatment. However, these results raise doubts on the need of broad-spectrum antibiotic use in patients with no observable need for antibiotic treatment as it may threaten the colonization process and subsequently alter infant development [105,106].

Finally, despite these differences in the bacterial colonization of full-term infants, preterm infants have been shown to be able to catch up to full-term infants through nutritional and probiotic interventions [107]. The optimization of infant nutrition allows for a catch-up period to be projected in order to allow preterm infants to achieve the growth and development parameters of full-term infants [108,109].

### 3.4. Neonatal Bacterial Colonization

Full-term infants have a specific path towards bacterial colonization distinct from that of preterm infants, which may be influenced by in utero colonization. The main determining factor in infant gut microbiota composition is the method of delivery.

#### 3.4.1. Delivery Method

Using the 16s rRNA sequencing of the neonatal skin, oral mucosa, nasopharyngeal aspirate, and meconium, Dominguez-Bello et al. were able to compare the microbiome of the neonate with the dermal, oral, and vaginal microbiome of the mother in relation to the delivery method. The results of this study indicate that the method of delivery determines the contents of vertical transmission, with neonates delivered vaginally taking on vaginal microbiota dominated by *Lactobacillus*, *Prevotella*, or *Sneathia* species and neonates delivered via C-section resembling the microbiome of the maternal dermis, dominated by *Staphylococcus*, *Corynebacterium*, and *Propionibacterium* species [110,111]. Put together, these observations support a variation in the sterile womb hypothesis: the bacterial baptism hypothesis. The assumption of this hypothesis is that bacterial colonization occurs during birth as a result of exposure to the bacterial composition of the vaginal canal or the dermis. Additional observation that supports this hypothesis is the finding that infants delivered through C-section are more prone to pathogenic bacterial infection than infants delivered vaginally, as the baptism in pathogenic dermal bacteria leads to increased concentrations of *Klebsiella*, *Enterococcus*, and *Clostridium* species (Figure 4) [112].

Despite the popularity of this hypothesis, researchers disagree about the importance of environmental factors when considering infant primary inoculation with bacteria. A twin study by Palmer et al. observed compositional differences in microbial profiles between dizygotic twins over the first six months of life, before stabilizing into an adult-like profile by the end of the first year of life [113]. These results suggest an environmentally dependent microbial composition after birth. This does not necessary counter the bacterial baptism hypothesis but calls into question the importance of initial inoculation.

Further complications arise when considering the importance of vertical transmission in cesarean delivery. Studies into the causal factor behind the discrepancy in bacterial colonization seen in infants born by C-section disagree. The main culprit may be considered the procedure itself, which seems to compromise the natural vertical transmission of bacterial mass through vaginal colonization and potential fecal exposure [114]. In children born via C-section, components of microbiota play a very prominent role in driving Treg differentiation, which is essential for controlling inflammation [115] and may contribute to common respiratory and neurological disorders such as autism spectrum disorder, schizophrenia, and immune diseases like asthma, skin atopy, juvenile arthritis, coeliac disease, and type 1 diabetes in the long term [116,117]. Stress and respiratory immaturity contribute to these issues, with cortisol levels being lower in C-section babies, leading to a reduced immune response as well. In particular, TNF-α and IL-6 responses to TLR1–2 stimulation are significantly reduced in C-section-delivered neonates compared to those delivered vaginally [116]. Elevated basophils and eosinophil and basophil progenitor cells in cord blood are also associated with a greater Th2 immune response, increasing the risk of respiratory symptoms, wheezing, and bronchitis in early infancy [118].

Studies show that respiratory tract infections are more common in babies born through C-section. Supplementing this, asthma can be observed to be more common in C-section children, suggesting a risk factor for asthma development. Additionally, C-section is associated with a higher likelihood of increased wheezing, hypersensitivity, dermatitis, atopy, obesity, and inflammatory bowel disease, with the effects on asthma and wheezing persisting up to age 18 [119,120]. Although cesarean delivery does not show a long-term impact on respiratory symptoms in the first year of life, it can later increase the risk of childhood asthma, allergy rhinitis, atopic dermatitis, respiratory tract infections, and obesity later in life [121,122].

However, some research proposes that the procedure plays a less active role in colonization. Instead, a shift to assessing the effect of intrapartum antibiotic administration, absence of labor, differences in breastfeeding behaviors, maternal obesity, and gestational age has provided evidence that a lack of vaginal exposure may not be the main reason for a different microbial phenotype [123]. Antibiotic use has been particularly studied in this regard, being linked to long-term disruptions of the natural microbiome, leading to developmental disruptions [114]. The results of these studies indicate a need for reevaluating the overall use of antibiotics in full-term deliveries when antibiotic use is not needed. Paired with the importance of environmental factors, these results propose a nurture-based view of microbial colonization, affected by drug exposure, nutrition, and post-natal care to establish the basis of an adult-like microbial phenotype.

#### 3.4.2. Long-Term Development of Gut Microbiota

The maturation of the infant gut microbiome occurs during the first years of life and is impacted by several environmental factors before reaching an adult-like composition. In their longitudinal study on the development of the gut microbiome in the first 2 years of life, Wernroth et al. conclude three broad findings about the change. Firstly, the infant gut microbiome is low in diversity and compositionally different between individuals. Secondly, through exposure to environmental factors such as maternal breastfeeding and nutrition, the perinatal factors of colonization, such as the mode of delivery, can be overcome in preference for a mature microbiome. Finally, neonatal oral and gut microbiota share similarities in the early months after birth but become distinct by the two-year mark [124]. Additional findings highlighting the importance of environmental factors come from a study by Barker-Tejeda et al., whose comparative analysis between the microbial composition of the infant gut and the microbiome of three generations of their ancestors showed that the microbiome of the infant is low in diversity and metabolite concentration, suggesting a strong environmental impact on the development of a mature microbiome [125].

The progression of microbial stabilization within the infant gut takes place over the early years of life. The neonatal microbiome is particularly rich in *Proteobacteria* species whose population gradually dwindles over the first year of life, reaching a concentration of <5% of overall gut biomass around the two-year mark, reflecting the composition of the adult gut microbiome [124]. The dominant anaerobic bacteria are discussed further below.

##### Bifidobacterium

*Bifidobacteria,* belonging to the phylum Actinomycetes, are Gram-positive anaerobic bacilli. Their concentrations within the infant gut oscillate during the first months of life, before finally reaching their stable concentrations at around 2 years of age [124]. Early peaks in *Bifidobacterial* concentrations can be explained by their central role in the processing and metabolism of prebiotic human milk oligosaccharides (HMOs), which direct the development of the infant gut microbiome during breastfeeding [126]. Additional evidence for their importance in healthy gut development comes from *Bifidobacterium* supplementation in preterm infants, leading to stabilization in bacterial composition [127].

##### Lactobacillacea

*Lactobacilli*, belonging to the phylum Firmicutes, experiences a universally significant increase in concentrations during early development before acquiring stable concentrations at 24 to 30 months [124]. As mentioned previously, *Lactobacilli* are a dominant species of the vaginal microbiome and are vertically transmitted during vaginal delivery onto the neonate. This is notable as infants delivered through cesarean section have a distinct lack of *Lactobacilli* colonization. An additional source of *Lactobacillacea* colonization is maternal milk during breastfeeding, which accounts for the continual increase in concentration from birth [128].

##### Clostridium

*Clostridia* also belong to the phylum Firmicutes; however, unlike *Lactobacilli*, they are mostly pathogenic. Despite this, the infant gut is particularly prone to colonization by this genus and presenting asymptomatically. The exact colonization pattern and importance of the genus is not fully understood; however, one study indicates that the colonization prevalence of this genus in infants can be as high as 27.2% and even higher in infants < 6 months of age [129]. The function of this colonization is not agreed upon but may be linked to pathogenic resistance, as the administration of *Clostridium* increases the colonization resistance of the infant gut and promotes homeostasis [130].

##### Bacteroides

As opposed to other phyla, *Bacteroides* experience a low change in diversity during infant development and seem rather to be implanted at a stable concentration during delivery [124]. The method of delivery is particularly important in this case, as infants delivered vaginally have an overall higher concentration in *Bacteroide* species after birth [131]. The function of this genus in the infant gut includes the fermentation of HMOs and synthetizing SCFAs for the promotion of healthy maturation. Additionally, some Bacteroide species, such as *B. fragilis*, have been found to be pathogenic and may play a role in the onset of allergies, as an increase in their concentration is seen in allergic infants [132].

#### 3.4.3. First 1000 Days of Life and Nutritional Weaning

During the early months of infant life, the primary source of nutrition is human breastmilk, which provides an opportunity for further colonization and exposure to prebiotic agents such as HMOs. Overall, the infant gut experiences a selective pressure, which decreases the concentrations of facultative anaerobes present shortly after birth (*Enterococcus*, *Staphylococcus*, *Streptococcus*, *and Enterobacteriaceae*) as the levels of free oxygen in the gut decrease. In the absence of oxygen, anaerobic species such as *Bifidobacteria*, *Lactobacilli*, and *Bacteroides* take hold [133].

Additional selection pressure comes from breast milk, containing secretory IgA as well as prebiotics, such as HMOs, which can be fermented by specialized bacterial strains, allowing for the diversity of *Bifidobacteria*, *Lactobacilli* and *Clostridium* species to increase. The role of HMOs is particularly important in this regard as their fermentation requires a particular enzymatic profile capable of degrading HMOs into monosaccharides [134]. The prevalence of these oligosaccharides in the nutrition of the infant promotes the growth of species capable of their fermentation and the addition of further metabolic pathways for the benefit of the host. Additionally, the increased concentrations of these bacteria in the gut provides a protective mechanism against colonization with potential pathogenic bacteria at this vital point in infant development (Figure 4) [134].

HMOs in breastmilk stimulate the outgrowth and diversification of Bifidobacteria as indicated above. Bifidobacteria have an immunostimulatory effect because they indirectly, via a mechanism called cross-feeding, stimulate the production of SCFAs [135]. In order to study the immunoregulatory effects of HMOs more directly, i.e., in the absence of other potential immunoregulatory components in breastmilk, the short-chain galacto-oligosaccharides (GOSs), long-chain fructo-oligosaccharides (FOSs), and 3-Fucosyllactose (3-FL) have been used most often. In at-risk children, supplementation with a combination of GOSs and FOSs significantly reduced plasma levels of total IgE as well as cow’s milk specific IgG1 [136]. In in vitro studies, combinations of butyrate and 3-FL in vitro reduces Th-2 cell frequency, increases Treg frequency, and increases the production of IFN-γ, IL-17, and IL-10 [137,138]. Overall, these data support the notion that HMOs can, via SCFA induction, reduce Th2 activity while promoting Th1 and Treg function. These effects are compatible with in vivo studies suggesting that HMOs can reduce the incidence of atopic dermatitis in infants at risk for allergic diseases [139], although these effects need to be substantiated.

Usually around 6 to 12 months after birth, the diet of the child shifts from exclusive breast milk to a solid diet, with dramatic changes in the infant gut microbiota as a consequence. During this transition (weaning), the overall concentrations of HMO-related bacteria such as *Bifidobacterium*, *Lactobacillus*, and *Enterobacter* decreases, while *Bacteroides*, *Akkermansia*, and *Ruminococcaceae* species expand as their enzymatic profiles, capable of processing broader classes of macromolecules, become beneficial and environmentally dictated [133].

At this point in infant development, the particular nutritional choices of the guardian(s) become the dominant factor in directing microbial composition. A study on the development of the natural metabolic profile of infants has found that the composition of the first complimentary food may be influential, as early nutrition lacking iron has been associated with inflammation and disruptions to the microbial profile of the infant [140].

The weaning from breast milk and the introduction of solid food thus has dramatic consequences for the composition of the gut microbiota. During weaning, the overall concentrations of HMO-related bacteria such as *Bifidobacterium*, *Lactobacillus*, and *Enterobacter* decreases, while *Bacteroides*, *Akkermansia*, and *Ruminococcaceae* species increase. In mouse models, it has been shown that weaning causes a temporary peak in TNF-α and IFN-γ. A delay in the introduction of solid food of less than 1 week (day 21 to day 27 post-partum) abrogates this spike in TNF-α and IFN-γ. The relevance of this weaning reaction only becomes evident in the adult life of the mouse: the delayed weaned mice are more susceptible to immune mediated diseases such as colitis [141]. Comparable data for humans are unknown, so the optimal timepoint of weaning from breastfeeding and the introduction of solid food, as well as potential changes in microbiota composition and functionality, will have to be determined. A multicenter study into the impact of the introduction of solid food and weaning on the colonization pattern of the infant gut is ongoing [142].

## 4. Cellular and Molecular Interactions Between Microbiota and the Immune System

As indicated and discussed above, the gut is the major interface between the microbiota and the developing immune system. It is estimated that 70–80% of immune cells reside in and around the gut, where they maintain a fine balance between anti- and pro-inflammatory responses [143]. The maintenance of intestinal homeostasis is highly dependent on the regulation of the immune system. As indicated above, microbe–host interactions during the neonatal period determine the immune mechanisms that maintain homeostasis and tolerance [144,145,146]. The GI tract and the mucosal immune system are highly interconnected, and it is important to understand the cellular and molecular interactions involved.

### 4.1. Gastrointestinal Microbe Activity

The GI tract is composed of a mucosal layer (exposed to the lumen), a single layer of epithelial cells, and the lamina propria [147]. The mucous layer represents a physical barrier between the intestinal epithelial and the microbiota which reside in the lumen [144,148]. Below the mucosal layer is the epithelial layer composed of tightly packed epithelial cells, whose major function is the absorption of nutrients such as sugars, vitamins, and amino acids [149]. Interspersed between the epithelial cells are immune cells such as T cells [144,147]. Below the epithelium is the lamina propria, a layer of connective tissue which supports the epithelium above and is rich in immune and other cells [144,147]. Finally, located within the lamina propria are Peyer’s patches, areas of lymphoid tissues which help monitor for, recognize, and process antigens [144,150].

Gut microbiota metabolizes proteins and carbohydrates thereby producing metabolites, such as vitamins, signaling molecules, and SCFAs, most notably butyrate, propionate, and acetate [145,151,152,153,154]. SCFAs act on many different cell types, through diverse mechanisms, to regulate important processes. They help with cell growth regulation, injury recovery, maintaining barrier integrity, and initiating, enhancing, limiting, and terminating immune responses [145,154,155]. They have also been shown to be involved in shaping certain structures of the immune system, such as Peyer’s patches, regulating amounts of immune cell types, and influencing the development of immune cells [143,156,157,158]. Commensal microbiota also adopts many mechanisms to prevent infection and the resulting spread and damage of pathogens. For instance, in a process termed ‘colonization resistance’, commensal microbiota makes the microbiome inhospitable to pathogens by altering pH and competing for nutrients, adhesion sites, and receptors [143,159]. They also secrete certain antimicrobial peptides (AMPs) such as α-defensins, RegIII, and lysozymes [144,160]. Additionally, bacteria self-regulate via quorum sensing, which is the ability to use secreted signaling molecules to monitor their environment and change gene expression in response to received signals, causing changes in motility, adherence, and population density [148,150,161]. Commensal bacteria use quorum sensing to maintain homeostasis, while pathogens can use it to limit immune response, among other activities. The interaction of microbiota, both commensal and pathogenic, with the GI tract results in continuous immune signaling, which is important for proper immune stimulation and development.

The neonatal immune system is immature, and both innate and adaptive immune responses are not fully functional (see above Section 2). Microbial interaction with the neonatal gut likely helps direct immune system maturation, playing a key role in developing a balanced innate and adaptive immune system [144,147,159]. Certain pioneer microbiota, such as *Bifidobacteria*, *Firmicutes*, and *Bacteroidetes*, are likely responsible for the early education of the neonatal immune system [159,162,163,164,165]. Proteobacteria are important for the early priming of innate and adaptive systems [146]. As mentioned previously, these anaerobic bacteria are also some of the most common and important colonizers in the neonatal gut. These bacteria contribute to the creation of a more desirable environment for beneficial microbial activity through mechanisms such as creating an anaerobic environment and colonization resistance [159,163,166]. Additionally, they produce and present specific compounds including microbe-associated molecular patterns (MAMPs), which help train immune tolerance [143,159]. The most important immune consequences of these bacteria involve the activation of T cells, B cells, immunoglobulins, and dendritic cells (DCs). These immune components, as well as the function of the mucosal and epithelial layers, will be discussed in further detail below.

#### 4.1.1. Interactions at Mucosal Surfaces

The mucosal immune system comes into direct contact with many external threats, especially from pathogenic microbes, so it has many structures and mechanisms to maintain immunological homeostasis and tolerance to environmental encounters. The mucosal layer is primarily composed of highly structured glycoproteins known as mucins, which provide an important source of nutrients for commensal bacteria [147,157,167]. Up to this point, 21 different mucin genes (Muc) have been identified [168]. Muc2 is the most expressed mucin in mucus [167,168]. It can bind and inhibit microbial translocation and is crucial for the proper functioning of the mucosal barrier [147]. The structure of the mucin network contributes to the viscoelasticity of mucus [169,170]. Other components such as proteins and DNA also contribute to the viscosity of mucus [167]. The mucus also acts as a storage for antimicrobial molecules, such as secretory immunoglobulin A (sIgA), defensins, and AMPs [167,171]. The mucus is produced and maintained by goblet cells located in the epithelium [147,167]. The mucus layer is a dynamic barrier and is able to change in thickness and coverage in response to immune interactions [147].

The mucus barrier is still immature at birth. The protective function of the neonatal mucus may be limited, as neonates have a thinner mucus layer (relative to adults) covering the epithelial layer due to the reduced expression of major mucins Muc2, Muc3, and Muc5ac [157]. The reduced thickness and amount of mucus is associated with increased permeability, which is then also associated with an increased potential for bacterial translocation [172]. Rat studies have shown increased susceptibility to certain infections in younger pups in comparison to older pups. The change in susceptibility was shown to be associated with the maturation and thickening of the mucosal layer, as well as the increase in mucin, DNA, and IgA amounts which impact viscosity [147,172]. One proposed explanation for reduced mucus production in early neonates is the reduced goblet cell density. This is supported by findings from pig studies [173]. Additionally, trefoil factor (peptides secreted by goblet cells) levels are significantly lower in preterm as compared to term infants, giving further support to the theory [147,174]. Trefoil factors are important for barrier restoration after injury, especially from damage caused by inflammation [175,176]. Therefore, it is possible that the immaturity of the neonatal mucosal barrier not only increases susceptibility to infections but also reduces the ability to recover from possible harm.

The mucus layer is one of the main GI areas impacted by microbiota [144]. It continuously comes into contact with microbes and is responsible for providing protection against pathogens while maintaining tolerance to commensal microbiota. It also allows for the segregation of microbiota and host immune cells due to its selective permeability, which helps to prevent abnormal immune responses [144]. The secretion and degradation of mucus are regulated by the host’s recognition of bacterial metabolites and MAMPs [143,177]. Mucus production and composition can also be altered by changes in inflammation and microbiota composition [159,178]. For example, SCFAs, which are the main metabolites produced by microbiota, can increase the secretion of mucus and AMPs from IECs [143,179]. The mucosal layer is the primary location of interaction between the microbiota and the immune system and serves an essential function in neonatal immunity.

#### 4.1.2. Epithelium

Despite the need for defense against potential harm from foreign substances, the gut epithelium consists of a single layer of cells only, connected by tight junctions [143,147,180]. Because of its location directly below the mucosal layer, it is exposed to the contents of the lumen. The epithelial barrier matures during gestation and is fully developed in full-term infants [157]. The development of the epithelial structure is a closely regulated process. The intestinal crypts and villi begin to develop in utero from intestinal stem cells (ISCs) between 8 and 24 weeks gestation [181,182]. During this period, the crypts deepen, the villi lengthen, and the ISCs proliferate and migrate to the crypt bases [181]. The ISCs also differentiate into many IEC subtypes, such as goblet cells and Paneth cells [182,183,184]. Differentiated cells migrate from the crypts up to the villi; however, Paneth cells and undifferentiated ISCs remain in the crypt bases [183].

The integrity of the epithelium is crucial for limiting foreign material, such as microorganisms and toxins, from entering the body. Interactions between gut microbiota and IECs can impact the permeability and function of the epithelial barrier [145,147,155,178,185]. For example, some pathogenic microbiota can release toxic metabolites that disrupt the formation of the tight junctions between IECs, reducing epithelial integrity and increasing permeability [180]. On the other hand, commensal SCFA-producing microbes improve barrier integrity by reducing inflammation and regulating cell turnover [145,159]. The continuous exposure of microbes in the gut causes adaptations in the neonatal gut tissue; however, this process is not yet fully understood [159]. The immature immune system of a neonate most likely plays a role in this, as it is characterized by dampened pro-inflammatory cytokine responses, resulting in a greater tolerance to microbiota establishment [159,186]. Tolerance is the main difference in function between neonatal and adult immune cells [162].

IECs can passively and actively recognize and take up SCFAs through passive diffusion and carrier-mediated absorption through solute transporters [154]. Transporters allow for the more efficient transport of SCFAs from the lumen of the gut into IECs, the lamina propria, and eventually the blood circulation. The primary transporters for SCFAs are SLC5a8 (sodium-coupled monocarboxylate transporter 1) and SLC16a1 (mono-carboxylate transporter 1) [154,187,188,189,190]. SCFAs are the most abundant microbial metabolite in the lumen [145,154]. They act as a significant source of energy to support cells when metabolized, as they can readily be converted into acetyl-CoA, fueling the tricarboxylic acid (TCA) cycle for ATP production [154,159,191]. They are also an important carbon source for IECs [145]. Additionally, SCFAs induce the proliferation of IECs and innate lymphoid cells, such as ILC3, which in turn produces IL-22, a cytokine which induces an anti-inflammatory response [154,179]. The metabolism of butyrate, a major SCFA, by IECs helps establish an anti-inflammatory environment [145,155]. Additionally, SCFAs can also help IECs produce AMPs, which help to limit epithelium and pathogen interaction [179]. Other metabolites besides SCFAs also have immunoregulatory properties. Bacterial tryptophan metabolites, such as iodine-3-propionic acid (IPA), impact the metabolism of tryptophan, an important amino acid that has been shown to improve epithelial integrity and help maintain the homeostasis of the mucosal immune system.

The regulation of immune responses is crucial for the maintenance of gut homeostasis. IECs direct appropriate responses to commensal and pathogenic microbiota by producing immune regulatory signals. For instance, Paneth cells, located in the crypts of Lieberkühn in the intestinal epithelium, are secretory cells that produce AMPs including α-defensins and lysozyme [144,147,167]. These cells first appear at about 12 weeks of gestation but do not fully mature until the postnatal period [192]. Preterm infants may have reduced AMP secretion due to reduced levels of Paneth cells compared to term infants [147]. Microbial metabolites help mediate communication between the epithelium and immune cells. For instance, commensal microbiota use signaling molecules to communicate between each other, as well as the epithelium [143]. Much like the mucosal layer, the epithelium provides a physical and chemical barrier, which is essential to maintaining homeostasis and normal functioning in the body.

#### 4.1.3. Introduction of Milk and Its Impact on the Gut Microbiota’s Immune System

By week 12, the fetal gut begins to form villi and crypts, and around this time, the fetus starts to ingest amniotic fluid through the mouth, coming into contact with external stimuli for the first time [193]. The fetal gut is believed to have a stable immune landscape from the second trimester until birth, though, as discussed above, it remains unclear if a microbiome is present before birth. Necrotizing enterocolitis in fetuses has been raised as an argument for the presence of bacteria in the fetal gut, as these bacteria could inflame mucosal cells, leading to apoptosis and necrotizing enterocolitis. This condition, characterized by a necrotizing bowel, can rapidly progress to severe illness and death [194].

HMOs have been shown to promote healthy gut bacteria, counteracting harmful microbes in the fetal gut. The weaning period is marked by a “weaning reaction”, which triggers the increased expression of genes involved in antimicrobial immunity, including Reg3 defensins, mucins, and CC and CXC chemokines in dams [36,141,195]. This reaction supports the “hygiene hypothesis” and posits that exposure to diverse, colonizing microbiota in the infant’s GI tract is crucial for the development of a transient microbiome in addition to proper immune system priming in infants to influence the onset of allergy and asthma later in life [141,196]. While there is strong evidence for this in mice, it is more challenging to confirm in humans. However, longitudinal stool sampling and in-depth immune profiling have revealed innate and adaptive immune responses during early and mid-infancy, likely induced by intestinal microbes settling in after exposure to breast milk and the mother’s skin, similar to the weaning reaction observed in mice [36,197,198,199].

The dysbiosis of the fetal microbiome has been shown to have long-term negative effects on an infant’s immune system, regardless of later colonization [200]. There is a window of opportunity for microbiome education and exposure to environmental cues, although the exact timing of this window remains unclear [163,201]. It is believed, however, that food-derived antigens play a key role in establishing a threshold for immune responses and future host–microbe interactions, directly and indirectly shaping gut microbiota [36,201,202,203,204].

Nutrition plays an important role in shaping neonatal mucosal immunity, as it introduces antigens to the infant for the first time, helping the immune system learn to build tolerance to non-self, foreign entities on the gut’s mucosal surface. Breast milk contains HMOs, which promote the growth of beneficial bacteria such as *Bifidobacterium*, creating a distinct microbiome in breastfed infants compared to those fed only with formula [36,205]. Infants exposed to breast milk have been shown to have increased microbial diversity and enhanced stabilization of the gut microbiome. This is believed to lay the foundation for a more robust and resilient immune system, influencing the acquisition of atopic diseases, obesity, and type 1 diabetes as they transition into adulthood [36,206,207,208].

#### 4.1.4. Introduction of Solid Food and Its Impact on the Gut Microbiota and the Immune System

The introduction of solid food into the diet of an infant sees a change in the rapid decline in *Bifidobacterium* species and an increase in *Bacteroides*, *Bilophila*, *Roseburia*, *Clostridium*, and *Anaerostipes* in the infant gut microbiome [36,166,209]. The composition of the microbiota (and hence, the production of SCFAs) shifts with the introduction of solid food, leading to an increased level of pro-inflammatory cytokines (TNF-α and IFN-γ) and the promotion of the differentiation of RORγt^+^ T-regs in the intestine [141].

Indirect interactions through maternal microbial metabolites can also influence the development of a fetus’s immune system, even before exposure to solid food, as studies have shown that limited exposure to *E. coli* during pregnancy can lead to changes in the fetal immune system, specifically affecting certain immune cells in the gut, like type III innate lymphoid cells (ILC3s) and mononuclear cells, if the fetus was indirectly exposed to them prenatally [210,211].

### 4.2. Interaction Between Microbiota, Including Their Metabolites and Cells and Molecules of the Immune System

Beyond the structural components of the epithelial and mucosal layer, there are also complex interactions of the many immune signals produced by the microbiota, IECs, and the immune cells which reside in the epithelial layer and lamina propria.

#### 4.2.1. MAIT Cells

Mucosal-associated invariant T (MAIT) cells are induced early during postnatal life by vitamin B-2 synthesizing commensal bacteria [212]. MAIT cells are derived from CD4^+^CD8^+^ thymocytes and are IL-23-dependent [212]. These cells respond to riboflavin metabolites produced by commensals and presented by MR1 MHC molecules. The interaction between MAIT cells and microbiota is essential for their development and function in tissue repair during adult life.

##### Anti-Inflammatory Effects of Bacterial Metabolites

Within the gut microbiome, the immune environment is primarily anti-inflammatory. This is especially important for neonates, who ingest an incredible amount of unfamiliar foreign material, including microbiota as well as food particles. Signaling by commensal bacteria promotes and maintains immune tolerance. For instance, bacterial metabolites, such as the SCFAs butyrate, propionate, and acetate, suppress inflammatory cytokine production in Th cells. They can increase IL-10 and decrease IL-6 secretion, promoting the differentiation of naïve T cells into T-regs and inhibiting differentiation into Th17 [145]. Butyrate also induces CTLs to produce granzymes and inhibits the activity of pro-inflammatory cytokine IL-17 [154,213]. Furthermore, SCFA signaling can also enhance the function of T-regs, as butyrate promotes the release of anti-inflammatory cytokine IL-10 from T-regs [144,145,159]. Additionally, butyrate and, to a lesser degree, propionate inhibit histone deacetylase (HDAC) activity, resulting in epigenetic gene alterations and increasing T-reg interaction with dendritic cells (DCs) [145,151,154,214]. Therefore, butyrate and other SCFA signaling increases the expression of the transcription factor Foxp3, which is associated with extrathymic T-reg production [145,155,159,214].

Because the repertoire of the antigen receptors on B- and T-lymphocytes is considered to be complete, the immune system has the capacity for the specific recognition of all possible antigens. By definition, this would then include self-antigens. In order to avoid an auto-immune attack of auto-reactive B- and T-lymphocytes, these cells should be eliminated or at least inactivated during development. For auto-reactive precursor T-lymphocytes, which differentiate in the thymus, auto-antigens are presented by medullary thymic epithelial cells (by virtue of *AIRE*, the autoimmune regulator). This gene plays a role in the promotion of self-tolerance by initiating the destruction of autoreactive T cells that respond to self-antigens. Thetis cells are a class of RORγt^+^ antigen-presenting cells present in the intestinal lymph nodes during a critical window in early life, coinciding with the wave of peripheral regulatory T cell differentiation [215]. Based on the expression of unique transcription factors, including Spi-B and PU.1, Thetis cells resemble medullary thymic epithelial cells. This suggests that they play a role in the induction of immunological tolerance [215]. This aspect will be further discussed below.

Regulatory T cells are immunosuppressive T cells which can directly and indirectly cause anti-inflammatory responses through a variety of mechanisms. Naïve T cells in fetuses are more likely to differentiate into T-regs than pro-inflammatory T cells [157,159]. Gut microbiota, such as *Bacteroides*, *Clostridium*, *Lactobacillus*, *Bifidobacterium*, and *Streptococcus*, impact T-reg proliferation and activity [159]. For instance, their SCFA metabolites promote T-reg acquisition [158,214,216,217]. They can prime T-reg production [143,157]. *Bacteroides fragilis* downregulates pro-inflammatory cytokine IL-8 and interferon gamma (IFN-γ), supporting the anti-inflammatory function of T-regs [159]. *B. fragilis* may also activate and enhance T-reg function by activating toll-like receptor 2 (TLR2) with their lipopolysaccharide surface polysaccharide A (PSA) [144,159].

Bacterial polysaccharides can have immunoregulatory effects and benefits and can also impact T-regs. In particular, PSA, a capsular carbohydrate with anti-inflammatory effects produced by *Bacteroides fragilis*, has been studied extensively in this respect [144,156,159,218]. In mouse models, *B. fragilis* has significant beneficial effects on the immune system, particularly in inflammatory bowel diseases [156,218,219]. These immunologic benefits result from its production of PSA [156,220]. PSA helps prevent inflammatory disease by inducing IL-10-producing CD4^+^ T cell proliferation, including T-regs, as shown with mice and in vitro studies [156]. These cells have immunosuppressive activities. *B. fragilis* and PSA can also suppress the expression of certain pro-inflammatory cytokines [144,218]. For example, PSA-producing *B. fragilis* has been shown to inhibit IL-23 release [156]. However, when *B. fragilis* is genetically modified to prevent the synthesis of PSA, this is not observed, indicating the immunosuppressive impacts derived from PSA [156]. PSA can inhibit the production of TNF-α and IL-17 by intestinal immune cells [156,220]. PSA is a zwitterionic polysaccharide (ZPS), a class of bacterial carbohydrates [218]. All identified ZPSs have immunoregulatory properties [221,222,223,224].

Indole-3-propionic acid (IPA) is another metabolite of tryptophan which significantly impacts immune function, especially related to T cells. IPA is a metabolite of tryptophan, produced by many microbiota [225]. Tryptophan is converted into indole-3-lactic acid (ILA) and then into IPA with the help of microbes, such as *Lactobacillus johnsonii* and *Clostridium sporogenes* (Figure 5). IPA regulates the stemness of CD8^+^ T cells [225]. IPA also promotes the proliferation of progenitor exhausted CD8^+^ T cells by the acetylation and modification of H3K27 [225]. Furthermore, IPA can inhibit inflammatory factors, including TNF-α, IL-1β, and IL-6 [226].

##### Pro-Inflammatory Effects of Microbiota

While signaling molecules from microbiota within the gut are primarily anti-inflammatory, they can still be recognized as antigens and trigger an inflammatory immune response. Most notably, intraepithelial cells, like TCRαβ^+^ and TCRγδ^+^ T cells, help protect IECs from injury and trigger immune responses by expressing cytokines [144,227,228,229]. Th cells secrete pro-inflammatory cytokines IL-22 and IL-17 in the lamina propria [143,144]. What do these cytokines do? Pathogenic microbiota, including segmented filamentous bacteria, can promote pro-inflammatory Th17 development [69,144,157,214]. In addition to the promotion of Th activation, signaling molecules can also activate and prime CTLs [144,155,225]. Butyrate inhibits HDAC in CTLs, which induces CTL differentiation into cytotoxic memory T cells and SCFA signals also prime CTLs [155].

#### 4.2.2. B Cells and Immunoglobulin A

Peyer’s patches along the GI tract consist largely of B cells. The production and activation of B cells causes the generation of plasma cells, which can express and produce serum IgA (sIgA) [144]. Germ-free mice have been shown to have reduced IgA-secreting plasma cells [230]. This indicates that gut microbiota drive IgA production. The release of sIgA is a major defensive mechanism of the mucosal immune system. It is responsible for blocking microbe adhesion to epithelial cells without causing inflammation [159,231]. In neonates and young children, a significant part of IgA is derived from breast milk [147,157].

Microbiota induce sIgA production when their antigens are taken up by microfold cells (M cells) found in the epithelial layer, causing B cell activation and the eventual increased production of IgA [144,147]. In addition, IgA may influence the microbiome by regulating the expansion of microbes by controlling their gene-expression, thereby impacting microbiome diversity. While part of adaptive immunity, IgAs have been shown to be polyreactive and to attach to a range of certain microbiota, indicating possible innate recognition.

#### 4.2.3. Dendritic Cells Connecting Gut Microbiota to Immune Systems

Dendritic cells (DCs) are important immune cells that link the innate and adaptive systems by expressing pattern recognition receptors (PRRs) and antigens [6]. Host–microbe interaction relies on the recognition of MAMPs using PRRs [143]. Bacteria are sensed differently depending on the type of PRR that interacts with their MAMPs [232,233]. Commensal microbiota are recognized by the mucosal immune system through MAMPs, which is important to induce tolerance [143,159]. This is also important for the healthy colonization of microbes. Fetal DCs contribute to immune regulation. They induce the differentiation of regulatory T cells (T-regs) and promote T cell IL-4 production, supporting a Th2 response. They also inhibit T cell TNF-α by increasing arginase 2 expression in conventional DCs within the fetal spleen to further aid in the suppression of inflammation [41]. Therefore, DCs are vital components of the mucosal immune system and are crucial for the initiation and control of immune responses.

These immune cells are considered immature at birth. Healthy neonates appear to express adult-level Toll-like receptors (TLRs) on DCs, an important PRR [144,159,234]. However, the production of innate inflammatory mediators is reduced in early life [159]. The response of TLRs to ligands expressed by microbes causes the release of chemokines and regulatory cytokines. Neonatal TLR signaling also promotes tolerogenic phenotypes of DCs, which results in immune responses to greatly involve T-regs [159]. Neonatal DCs promote the differentiation of Foxp3^+^ T-regs and limit TNF-α production by inducing the proliferation of allogeneic T cells [41]. Further induction of T-regs occurs due to C-type lectin receptors on DCs, which recognize and bind to certain microbiota [159,235]. Furthermore, DCs also secrete anti-inflammatory cytokines. IFN-β DCs, which release IFN-I, can be induced by the glycolipids released by *B. fragilis* [236].

## 5. Conclusions

The intimate and intricate interaction between gut microbiota and the neonatal immune system is crucial for the regulated development of immunity: efficient and adequate defense against pathogens while maintaining tolerance for commensal microbiota. *Homo sapiens* and their microbiota function as a superorganism with mutual benefits: the intestines offer safe niches for a great diversity of micro-organisms, which in turn contribute to the health and well-being of their host by supporting the immune system, as well as providing vitamins and other metabolites. When *Homo sapiens* and their microbiota function as one, the immune system must consider the antigens expressed by commensal microbiota as self. The induction of self-tolerance takes place in the thymus and bone marrow during the maturation of T cells and B cells, respectively. Microbial antigens are not expressed at those sites, so tolerance induction necessarily has to take place at other sites, of which mucosal surfaces are the most likely. As discussed in Section 2.1.4, during the intrauterine period, the fetus can swallow amniotic fluid, and this fluid contains microbial antigens from the mother [41,177,234]. Via such a mechanism, the fetal immune system could become tolerant for maternal antigens, including maternal microbiota. The second, and most dramatic, wave of exposure to the outside microbial world takes place during and immediately after birth (in the timeline of Figure 6, day 0). During passage through the birth canal, the neonate comes into direct and intense contact with maternal microbiota (Section 3.4). The activation of pattern recognition receptors, including the ones expressed on buccal cells, initially triggers a strong inflammatory response, but within days, this response wanes [237]. In the first weeks after birth, a rapid succession of bacterial species in the infant gut takes place, with some species appearing and taking hold, while others can be detected for just one or several days and then disappear [238]. A third wave of exposure of the immature mucosal immune system to microbial antigens takes place during the introduction of solid food. At least in mice, this is accompanied with big changes in the composition of gut microbiota, as well as the exposure of the immune system to a wide array of food antigens. At the same period, Thetis cells accumulate in the lamina propria [176]. Thetis cells, which have features of both dendritic cells as well as ILCs, induce peripheral Treg cells. The current concept is that Thetis cells in such a way fulfill a role in the induction of tolerance for non-self antigens, particularly commensal bacteria and food antigens. Little is known yet about Thetis cells in humans, including the timing of their appearance and the exact role in the development of tolerance. Still, it is tempting to postulate that these cells play an important role in the establishment of the tolerance of the immune system to commensal bacteria. After the introduction of solid food in the diet and weaning from breast feeding, the diversity of gut microbiota progresses at a slower rate and, by day 1000 of life, reaches a stable plateau. Apart from major and permanent changes in diet and antibiotics, the composition of the gut microbiota remains stable well into adulthood.

A consequence of the above integration of microbiological and immunological data is that the functionality of the immune system determines the composition and diversity of gut microbiota. Because the fine specificity of the immune system is individually determined (being the consequence of rearrangements of immunoglobulin and T cell receptor gene segments), the composition of gut microbiota is also individually determined. Another implication would be that a permanent change in gut microbiota composition would not be possible, apart from drastic measures such as permanent dietary changes.

Key unresolved issues include the concept of in utero bacterial colonization. This is currently controversial, and better designed experiments need to be performed. Finding the optimal timing of weaning and the introduction of solid food in infants to support immune development is crucial. Finally, a deeper understanding of the cellular and molecular interactions between microbiota and the immune system would allow for targeted interventions in cases where the immune system would function super- or sub-optimally.

## Figures and Tables

**Figure 1 biology-14-00299-f001:**
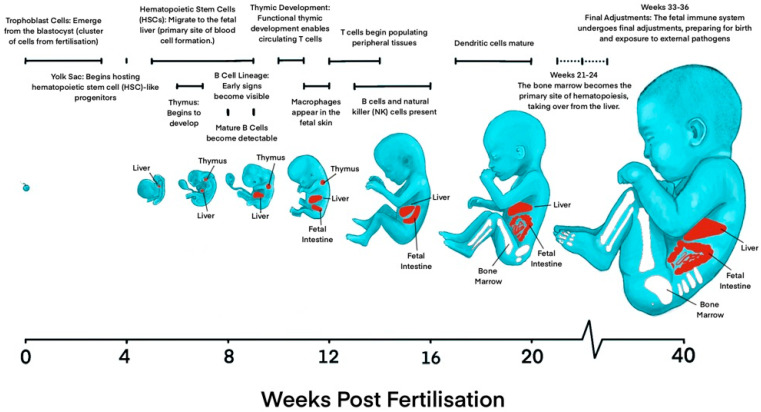
Ontogeny of the fetal immune system during prenatal development. Key steps in the development of cells and organs of the innate and adaptive immune system are indicated. See text for further details.

**Figure 2 biology-14-00299-f002:**
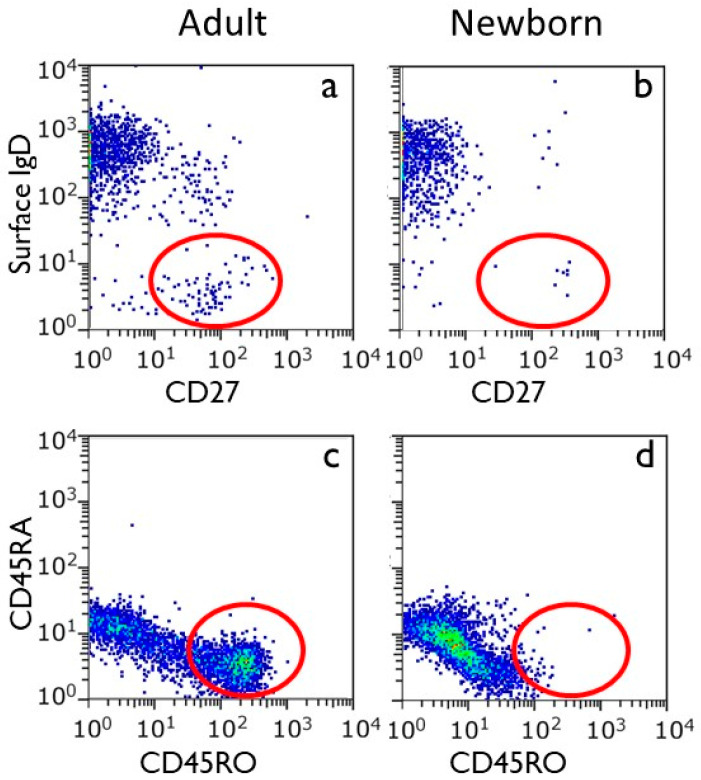
Naïve B- and T-lymphocytes in umbilical cord blood. In the upper panels, blood mononuclear cells were stained with CD19, IgD, and CD27 antibodies. B-lymphocytes were gated on CD19, and expression of surface IgD was plotted against CD27. In adult peripheral blood (panel (**a**)), memory B-lymphocytes, characterized by the phenotype surface IgD^−^ CD27^+^, are enclosed in the red circle. Memory B-lymphocytes in cord blood are virtually absent (panel (**b**)). In the lower panels, mononuclear cells were stained with CD3, CD45RA, and CD45RO. T-lymphocytes were gated on CD3, and memory T-lymphocytes, characterized by the phenotype CD45RA^−^ CD45RO^+^, are abundant in adult blood (panel (**c**)) and almost absent in cord blood (panel (**d**)). Data from GT Rijkers, unpublished.

**Figure 3 biology-14-00299-f003:**
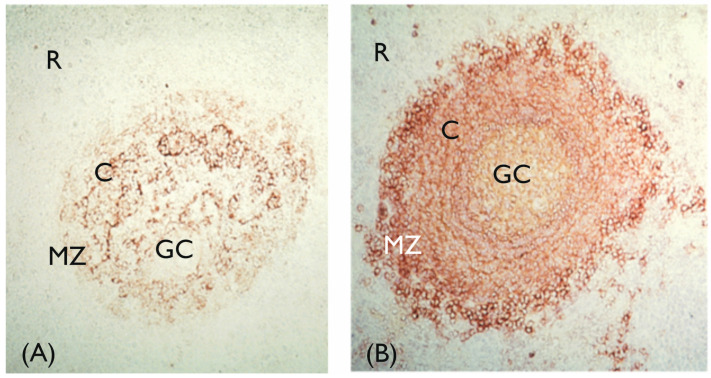
Expression of CD21 on human marginal zone B cells. Immunohistochemistry of the HB5 (anti-CD21) antibody on sections of a spleen of a 12-month-old infant (panel (**A**)) and a 13-year-old child (panel (**B**)). In panel (**B**), the marginal zone (MZ) B cells show strong expression of CD21, but the infant MZ B cells (panel (**A**)) are lacking CD21 expression. GC, germinal center; C, corona; R, red pulp. Courtesy of Prof. W. Timens. The photograph in panel (**B**) has been published before [55].

**Figure 4 biology-14-00299-f004:**
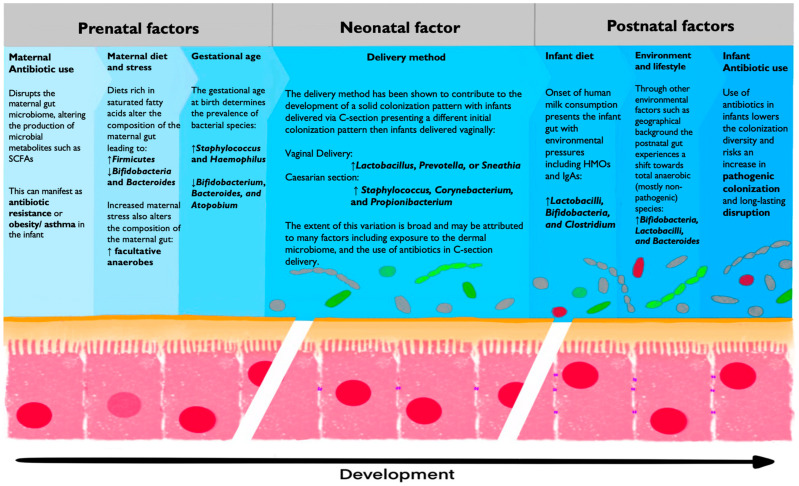
Major developmental factors in microbial colonization. The progressive colonization of the infant gut is shown under the categorizations of natal status. From left to right, the figure depicts the progressive colonization of the gut with commensal (gray), probiotic (green), or pathogenic (red) bacteria. The progression of tight junction maturation is also presented (purple). Up arrow indicates increase of relative abundance, down arrow decrease of relative abundance.

**Figure 5 biology-14-00299-f005:**
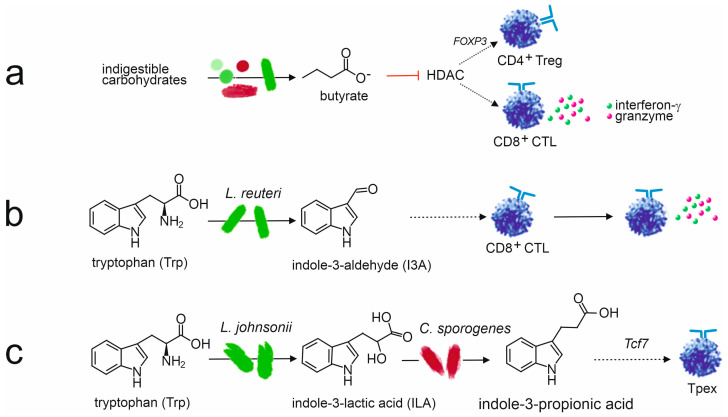
Microbiota metabolites with an impact on the functionality of the immune system. Panel (**a**) show that various bacteria are able to produce the short-chain fatty acid butyrate as a byproduct of the fermentation of indigestible carbohydrates. Butyrate inhibits histone deacetylase (HDAC) which leads to the activation of *FOXP3* in regulatory T cells. Butyrate also activates CD8^+^ cytotoxic T cells which produce IFN-γ and granzymes. Dietary tryptophan can be metabolized by *L. reuteri* into indole-3-aldehyde (I3A) (panel (**b**)) or by *L. johnsonii* and subsequently *C. sporogenes* into indole-3-propionic acid (IPA) (panel (**c**)). I3A, via unknown mechanisms, activates CD8^+^ cytotoxic T cells, while IPA leads to the activation of the Tcf7 transcription factor, resulting in the expansion of progenitor exhausted CD8^+^ cells (Tpex). See text for further explanation.

**Figure 6 biology-14-00299-f006:**
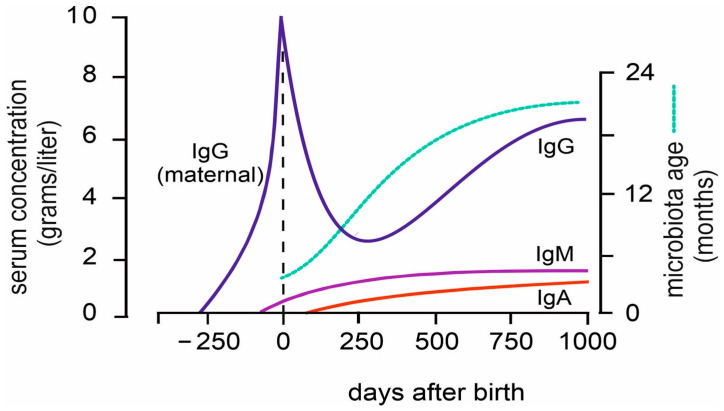
Immunoglobulin levels and gut microbiome development in early human life: IgG, IgM, and IgA levels are shown across fetal development, birth, and the first 1000 days postnatally. During the fetal period, IgG levels rise due to maternal transfer via the placenta. Postnatally, IgG levels initially decline as maternal antibodies are metabolized, but they gradually increase as the infant’s immune system matures and begins endogenous IgG production. IgM and IgA levels are present in smaller amounts, with both rising at a slower rate compared to IgG. The line representing bacterial diversity and abundance in the infant gut microbiome demonstrates a rapid postnatal expansion. This increase begins with early colonization and continues through the first 1000 days.

## Data Availability

Data sharing is not applicable. No new data were created or used for this review.

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
