# Peer review of "Mutual Interactions Between Microbiota and the Human Immune System During the First 1000 Days of Life"

_biology, 2025, doi:10.3390/biology14030299_

Round 1
Reviewer 1 Report
Comments and Suggestions for Authors
The manuscript authored by Tang et al. provides a thorough and well-researched review of the dynamic relationship between microbiota and immune system development in early life. The paper does a great job of covering both prenatal and postnatal immune system maturation, the influence of microbial colonization, and the intricate interactions between microbiota and immune cells and molecules.
The strength of this review lies in its comprehensive approach—it weaves together insights from immunology, microbiology, and developmental biology to create a clear picture of how the microbiota-immune system interplay unfolds in early life. The paper is well-structured, logically progressing from in-utero immune development to postnatal microbial colonization, all while incorporating relevant molecular and cellular details.
However, the reviewer has one concern for the authors’ consideration. At this time, the evidence for true prenatal bacterial colonization is highly contested. Although authors tried to cite papers from both sides, this reviewer believes that current weight of evidence suggests that the healthy human fetus likely does not harbor an established live bacterial community in utero.
Reviewer 2 Report
Comments and Suggestions for Authors
Tang et al. comprehensively review the interactions between microbiota and the human immune system during the first 1,000 days of life. This topic is highly relevant and of great interest in the fields of immunology, microbiology, and early-life development. However, several points need clarification, further elaboration, or more evidence to support affirmation. The following are specific questions and suggestions for improvement.
- Although the topic is debatable, the manuscript presents both the sterile womb and in-utero colonization hypotheses; however, it does not provide a conclusive evaluation of the evidence.
- The manuscript cites bacterial metabolites like SCFAs, indoles, and polysaccharide A, but it fails to detail their specific immunological effects. Please elaborate on that discussion.
- It is established that the mode of delivery impacts the offspring’s immune system. The authors describe differences in microbiota between vaginal and C-section deliveries but do not address long-term immune outcomes. Please include a discussion on that.
- The impact of maternal diet on fetal immune development is mentioned briefly but lacks detail. For example, how do maternal gut microbiota-derived SCFAs influence fetal immune priming via placental transfer? Please elaborate on this topic.
- Breastfeeding and its effect on newborn health and development is highly significant. Although the study mentions the importance of HMOs, it lacks a thorough discussion of their immunomodulatory effects.
- It is interesting that the "weaning reaction" is noted; however, supporting evidence is limited. The authors should provide more evidence regarding the immune transition during weaning and the optimal timing for introducing solid foods to support immune development.
- A summary of key unresolved questions would enhance the impact of the review.
- Figures are informative but referenced inconsistently in the text. Ensure all figures are cited appropriately and consider adding schematics illustrating microbiota-immune interactions at different stages of infancy.
- Figures 1 and 4, while informative and significant, appear quite dark, making it difficult to decipher the text within them. Please consider adjusting the color scheme, and do not use cursive.
- Some discussions lack proper citations, particularly in the sections on maternal microbiota and immune memory.
